

# Normalized group activations based feature extraction technique using heterogeneous data for Alzheimer's disease classification

Krishnakumar Vaithianathan[1], Julian Benadit Pernabas[2], Latha Parthiban[3], Mamoon Rashid[4], Sultan S. Alshamrani[5] and for the Alzheimer's Disease Neuroimaging Initiative[*]

[1] Department of Computer Engineering, Karaikal Polytechnic College, Varichikudy, Karaikal, Puducherry, India
[2] Department of Computer Science and Engineering, CHRIST (Deemed to be University), Kengeri Campus, Karnataka, India
[3] Department of Computer Science and Engineering, Community College, Pondicherry University, Puducherry, India
[4] School of Information Communication and Technology, Bahrain Polytechnic, Isa Town, Kingdom of Bahrain
[5] Department of Information Technology, College of Computers and Information Technology, Taif University, Taif, Saudi Arabia
[*] Data used in preparation of this article were obtained from the Alzheimer's Disease Neuroimaging Initiative (ADNI) database (adni.loni.usc.edu). As such, the investigators within the ADNI contributed to the design and implementation of ADNI and/or provided data but did not participate in analysis or writing of this report. A complete listing of ADNI investigators can be found at: http://adni.loni.usc.edu/wp-content/uploads/how_to_apply/ADNI_Acknowledgement_List.pdf.

Corresponding authors
Krishnakumar Vaithianathan,
vkichu77@gmail.com
Mamoon Rashid,
mamoon873@gmail.com

## ABSTRACT

Several deep learning networks are developed to identify the complex atrophic patterns of Alzheimer's disease (AD). Among various activation functions used in deep neural networks, the rectifier linear unit is the most used one. Even though these functions are analyzed individually, group activations and their interpretations are still not explored for neuroimaging analysis. In this study, a unique feature extraction technique based on normalized group activations that can be applied to both structural MRI and resting-state-fMRI (rs-fMRI) is proposed. This method is split into two phases: multi-trait condensed feature extraction networks and regional association networks. The initial phase involves extracting features from various brain regions using different multi-layered convolutional networks. Then, multiple regional association networks with normalized group activations for all the regional pairs are trained and the output of these networks is given as input to a classifier. To provide an unbiased estimate, an automated diagnosis system equipped with the proposed feature extraction is designed and analyzed on multi-cohort Alzheimer's Disease Neuroimaging Initiative (ADNI) data to predict multi-stages of AD. This system is also trained/tested on heterogeneous features such as non-transformed features, curvelets, wavelets, shearlets, textures, and scattering operators. Baseline scans of 185 rs-fMRIs and 1442 MRIs from ADNI-1, ADNI-2, and ADNI-GO datasets are used for validation. For MCI (mild cognitive impairment) classifications, there is an increase of 1–4% in performance. The outcome demonstrates the good discriminatory behaviour of the proposed features and its efficiency on rs-fMRI time-series and MRI data to classify multiple stages of AD.

## INTRODUCTION

Alzheimer's disease (AD) (*Alzheimers Association, 2024*; *Hashmi et al., 2022*) is a neurodegenerative disease and is one of the major causes of mortality all over the world. Although the internal damages caused by AD are extensive, it mainly targets memory. The reduction in brain mass, posterior atrophies in cortical regions, degeneration of brain tissues in frontal, temporal, and parietal regions, and enlargement of ventricles over a period of time have a detrimental effect on the subjects' speaking and memory capabilities (*Bailey-Taylor et al., 2022*). Researches show that it took around 2–6 years to transform from mild cognitive impairment (MCI) to critical AD (*Aqeel et al., 2022*). A steady loss of subjects' focus/concentration, bad judgment in self-decision-making, exhibit uncertainty with time/place, and the onset of loss of memory are some of the major symptoms shown by these patients (*Gillis et al., 2019*).

It is impossible to overestimate the significance of brain atrophy index and AD, especially in light of aging-related public health concerns. AD is predicted to become much more common as the world's population ages, which makes it a serious public health concern. For a number of reasons, early detection and monitoring of AD progression are essential (*Woźniak et al., 2024*). First of all, prompt intervention can greatly enhance patient outcomes, and it is made possible by early detection. Although AD now has no known cure, there are therapies that can help control symptoms and delay the illness's course. But when these therapies are started early in the course of the illness, they work best. As a result, applying brain atrophy indices can help detect AD early and enable timely management.

Early detection/monitoring can also lower the cost of AD-related medical care which has a significant financial impact, including indirect costs like missed productivity and carer stress in addition to direct hospital costs. Healthcare professionals can initiate therapies targeted at slowing the advancement of AD and lowering the requirement for expensive long-term care by detecting the disease in its early stages. Through precise identification of individuals at risk for AD and ongoing monitoring of the disease's progression, healthcare practitioners can optimize resource allocation and guarantee that patients receive the necessary amount of care and support. Estimates of the yearly cost of treating and diagnosing AD go into the hundreds of billions of dollars worldwide (*Gustavsson et al., 2023*). This covers costs for prescription drugs, long-term-care services, medical consultations, and diagnostic testing. Healthcare stakeholders can make well-informed decisions about resource allocation and prioritise interventions that have the greatest potential to reduce the overall burden of AD on society by using models to analyse the cost-effectiveness of early diagnosis and intervention strategies.

Recently, there has been a significant increase in the use of machine learning and pattern classification based feature extraction techniques to evaluate the extent of neurodegeneration in AD (*Sharma & Mandal, 2022*). For MRI and rs-fMRI, deep

convolutional neural network is the most used deep learning model for AD detection (*Abbas, Chi & Chen, 2023*). Multimodal feature selection (*Shi et al., 2022*) uses different biomarkers from MRI and FDG-PET to improve the efficiency of AD classification. The AD biomarker analysis (*Khatri & Kwon, 2022a*) using multi-measures of structural MRI and rs-fMRI enhances the efficiency in separating AD from healthy subjects. Graph convolutional networks efficiently integrate the weighted clustering weights for fMRI-based function connectivities to perform AD analysis (*Lei et al., 2023*). However, the features extracted by these methods are specific to their modalities, *i.e.*, they cannot be applied to both MRI/rs-fMRI. Explainabilty of the features/models is another issue that is equally important as their performance in AD classification.

Among various activation functions used in deep neural networks for classification, the rectifier linear unit is the most used one. Even though these functions are investigated individually, the group activations are not explored in neuroimaging (*Sharif et al., 2022*). Group activations (*Klabjan & Harmon, 2019*) are tested and found to improve the accuracy of ISOLET, MNIST, STL-10, and CIFAR-100 datasets. There are no work on their interpretability. We intend to analyze the performance and explanation capability of the group activations for AD classification using both MRI and resting-state MRI. The contributions of this study are:

1. A novel feature extraction method based on normalized group activations is proposed for AD classification. It can be used for both structural MRI and resting-State-fMRI.
2. An automated diagnosis system with the proposed features using seven heterogeneous features is designed and analyzed on multi-cohort data (ADNI-1, ADNI-2, and ADNI-GO).
3. 185 Initial/baseline rs-fMRIs are taken from ADNI-2 and ADNI-GO datasets.
4. 1,442 baseline MRIs are chosen from ADNI-1 and ADNI-2. Performance is evaluated on seven heterogeneous features:
   a) ADNI-1 as training data and ADNI-2 as test data.
   b) ADNI-2 as training data and ADNI-1 as test data.
5. Unbiased evaluation of performance on rs-fMRIs and MRIs using seven heterogeneous features with repeated 10-fold cross-validation.
6. Data visualization of functional-connectivity/class-specific rollout-attention explainability using rs-fMRI/MRI.

At the heart of our proposed model is a new technique of generating features called normalized group activations, which is different from traditional deep neural networks that usually use monolithic activation functions (such as ReLU). The use of several normalizations generates features that are aggregated across many neurons and from various parts of the brain. This means that the model will be able to relate the function of all synapses into a single general interpretation of the brain activity. This method allows the model to be more sensitive to subtle patterns in the data, which is crucial for detecting Alzheimer's disease. Our model is different from a minimalistic one that isolate the actions

of individual brain regions, by capturing how the brain goes from subcortical to interconnected regions and routes during a certain task. This is particularly important in neurodegenerative diseases like AD, as regional interactions often deteriorate with individual regional decline.

The other significant improvement of our model is its versatility in use not only with the structural MRI but also with the resting-state fMRI data, hence providing a more comprehensive multimodular approach. The connection of the two modalities through the model's algorithms reveals causal representations by means of the MRI and the full connectivity network by the fMRIs respectively resulting in accurate predictions and enhanced knowledge of Alzheimer's disease. Usually existing single-modal models tend to process one of these modalitites, but our idea that merges them brings out the big picture in brain circut malfunctioning.

One of the major contributions of the model is the usage of Regional Association Networks which examine relationships of two brain regions rather than isolated regions. This process, which is supported by normalized group activations can detect diseased areas that are disperse and not homogeneously accessible. The normalization layer distributes the activation across all feature layers, thus, it does not only make the feature representation more interpretable but it also improves the classification accuracy. In summary, our model not only helps in automating the process of feature extraction, but also facilitates classification at different stages of AD through the use of diverse data inputs (for example, wavelets, textures) and provides a robust tool in real-world settings.

The structure of this article is as follows: "Data, Transformations, and Methods" describes the datasets, preprocessing pipeline, and mathematical transformation functions. "Normalized Group Activations Based Feature Extraction" explains the normalized group activations feature extraction technique proposed in this study. "Automated AD Diagnosis System using Heterogeneous Features ($T_f^{NGA-RA}$)" discusses the configuration of the AD diagnosis system using heterogeneous features and "Experimental Setup and Results" presents the experimental setup and results of seven different types of classifications using rs-fMRI and MRI datasets. "Discussion" presents the data visualization of functional-connectivity/class-specific rollout-attention explainability and compares the proposed model with SOTA 3-D deep learning methods. Then, the conclusion is given in "Conclusion".

## DATA, TRANSFORMATIONS, AND METHODS

### ADNI dataset

The dataset used in this study is taken from ADNI (Alzheimer's disease Neuroimaging Initiative). As a public-private collaboration, Michael W. Weiner. MD (the principal investigator), established ADNI in 2003. The major objective is to determine whether multiple modalities (MRI, fMRI, PET), clinical/neuropsychological evaluation measures, and various other bio-indicators can be combined to monitor the progression of moderate cognitive impairment and early AD. Table 1 summarizes the demographic details of both ADNI-1, ADNI-2, and ADNI-GO datasets. The subject scans are downloaded in the NIfTI

**Table 1 Demographic data of all the subjects used in this study.**

|  | Type | No. | Age | Gender | MMSE |
|---|---|---|---|---|---|
| **rs-fMRI** |  |  |  |  |  |
| ADNI-2 | NC | 44 | 75.1 ± 6.6 | 20/24 | 28.9 ± 1.2 |
|  | AD | 30 | 72.7 ± 6.5 | 15/15 | 22.7 ± 2.5 |
|  | eMCI | 50 | 71.7 ± 7.2 | 19/31 | 28.2 ± 1.5 |
|  | lMCI | 34 | 71.3 ± 8.4 | 21/13 | 27.8 ± 1.9 |
| ADNI-GO | NC | 8 | 74.9 ± 8.3 | 2/6 | 29.4 ± 1.1 |
|  | AD | 4 | 71.3 ± 12.0 | 1/3 | 23.0 ± 2.4 |
|  | eMCI | 9 | 75.9 ± 5.7 | 3/3 | 27.8 ± 1.0 |
|  | lMCI | 6 | 71.3 ± 4.4 | 3/6 | 28.3 ± 2.5 |
| **MRI** |  |  |  |  |  |
| ADNI-1 | NC | 229 | 76.2 ± 5.62 | 127/102 | 29.1 ± 1.0 |
|  | AD | 199 | 75.3 ± 7.5 | 106/93 | 23.3 ± 2.0 |
|  | MCI | 398 | 75.9 ± 7.6 | 256/142 | 27.0 ± 2.07 |
|  | cMCI | 167 | 74.8 ± 6.8 | 102/65 | 26.6 ± 1.7 |
|  | MCInc | 231 | 74.9 ± 7.68 | 154/77 | 27.3 ± 1.8 |
| ADNI-2 | NC | 188 | 73.9 ± 6.6 | 90/98 | 26.1 ± 1.7 |
|  | AD | 151 | 74.4 ± 7.9 | 88/63 | 23.3 ± 2.0 |
|  | MCI | 277 | 73.6 ± 8.0 | 158/119 | 27.8 ± 2.4 |
|  | cMCI | 38 | 71.3 ± 7.3 | 24/14 | 26.9 ± 1.7 |
|  | MCInc | 239 | 71.8 ± 7.58 | 134/105 | 28.4 ± 1.7 |

**Note:**
The values are denoted as mean ± standard deviation.

(neuroimaging informatics technology initiative) file format from the online ADNI data repository.

### rs-fMRI data and pre-processing

The rs-fMRIs are chosen based on the following search criteria on ADNI advanced beta search: Modality = *fMRI* AND Description = *Resting State fMRI* AND Type = *Original*. This search yields a total of 770 scans with 118 AD, 197 CN, 239 $e_{MCI}$, 159 $l_{MCI}$, 58 other types (including types MCI, Patient, and Significant Memory Concern (SMC). It contains images from two protocols ADNI-2 and ADNI-GO. For this study, only initial scans of all the subjects are considered, so a total of 185 rs-fMRIs with 34 AD, 52 NC, 59 $e_{MCI}$, 40 $l_{MCI}$ are selected for this work.

Here, an exhaustive seven-stage preprocessing pipeline is used. The first stage is the removal of skull and neck voxels which are non-useful regions detected from T1-weighted structural image data with respect to each time-course of rs-fMRI. FSL-BET (*Smith, 2002*) software is used at this initial stage. Then, the FST-MCFLIRT (*Jenkinson et al., 2002*), and Hanning-Windowed Sinc Interpolation (HWSI) correction are done. Each fMRI time-series is moved by a minimum fraction with respect to the center of the TR time. And a gaussian kernel of 5 mm full width half maximum is used to smooth it and a high-pass filter (sigma cut-off of 90s) is used to remove unwanted low frequencies. The fMRI slices are registered to the corresponding structural T1-scan with seven degrees of freedom

(affine transformation). Then, they are registered to the MNI152 (Montreal-Neurological-Institute standard brain template). Finally, the resampling of the aligned fMRI brain slices is done by a $4 \times 4 \times 4$ mm kernel and mapped to the standard AAL template with 116 regions. The above-preprocessing steps generate a 4-dimensional (4D) data array with a time-series of $T \, \varepsilon \, 124200$ (set of 140 data points per subject) is generated.

### MRI data and pre-processing

A total of 1,442 images produced from two protocols of ADNI are used for this study. ADNI-1 harmonized protocol (*Jack et al., 2008*) is used in the acquisition of T1-weighted scans. A total of 826 T1 baseline scans are selected for training purposes. This set comprises 199 AD, 229 NC (normal-subjects), 167 cMCI (conversion to AD from MCI in a span of 36 months), 231 MCInc (not-converted to AD from MCI in a span of 36 months). Around 616 subjects selected from the ADNI-2 protocol are used for testing. This set also includes 3T T1 weighted MR images and has 188 NC, 151 AD, 38 cMCI, and 239 MCInc subjects for comparing the performances of different deep learning models.

A multi-step pipeline is used to pre-process the structural MRI of the subjects. At first, the unusable regions like the skull and neck regions are removed from the scans (*Smith, 2002*). Then, the FSL-VBM (*Smith et al., 2004*) is used to segment the brain into GM (grey matter), WM (white matter), and CSF (cerebrospinal fluid). These segmented regions are combined to obtain the true pixels of the MRI. Finally, the combined regions in the previous step are linearly registered to the standard AAL template with 116 regions.

## Transformations

### New-texture-extraction model (RROI)

Gray-level co-occurrence matrices (GLCM) are typically used to derive texture descriptors. Also, textures can be calculated separately from various block sizes on multiple axes (axial, sagittal, and coronal). Haralick textures like central moments, contrast, entropy, inverse difference moment, and homogeneity are computed for each block and used for training. This is our previous work (*Vaithinathan, Parthiban & Initiative, 2019*). Let $I_s$ denote every slice image of size $K \times N$. The image feature extracted $I_s$ is split into $B_l \times B_l$ blocks and this can be written as:

$$I = \bigcup_{\substack{i=1..N_{b_l} \\ j=1..N_{b_l}}} B_{l_{ij}} \tag{1}$$

where $B_{l_{ts}} = \bigcup_{(k,l)} P_{k,l}$ such that $(N_{b_l}s + 1) \leq k \leq N_{b_l}(t + 1)$ and $(N_{b_l}t + 1) \leq l \leq N_{b_l}$ $(s + 1)$ and $P$ denotes the intensities of the image. The values of $B_l$ considered in this study are 15 and 21.

### GLCM textures

In MRI scans, imperciptible tumors and defective regions can be effectively quantized using textures. A total of 11 Haralick textures are computed for this transformation. In 3D-GLCM space, individual textures are computed in 13 dissimilar angles and the distances range is between 1 and 3 are used. For every block, 429 textures are computed.

For 3-D matrix of size $h \times w \times k$, thirteen orientations are used (*Philips et al., 2008*). It is denoted by:

$$C_{\delta_p}(x, y) = \sum_{a=1,1,1}^{h,w,k} \begin{cases} 1, & \text{if } M(p) = x \text{ and} \\ & M(p + \delta_p) = y \\ 0, & \text{otherwise} \end{cases} \tag{2}$$

where $M(p)$ denotes the gray-level of the image at pixel $p$ and $M(p + \delta_p)$ is the pixel intensity at offset by $\delta_p$.

### 3D wavelets ($W_T$)

The decomposition technique of the 3D-discrete wavelet transform (*Katunin, Dańczak & Kostka, 2015*) can be obtained by performing tensor product of scaling and wavelet basis functional vectors. Both the vectors should be orthonormal, semi-orthogonal, bi-orthogonal, and have vanishing moments and compact support. There are eight basis functions in the tensor product of the vectors with the scaling function $\Phi_{a,b,c,d}^0(x, y, z)$ and seven orientational wavelet basis $\Psi_{a,b,c,d}^\tau(x, y, z)$ where $\tau = 1$ to 7. The decomposition of the 3-D signal $f(j, k, l)$ is given as:

$$f(j, k, l) = \sum_b \sum_c \sum_d h_{b,C,d}^A \Phi_{A,b,c,d}(j, k, l)$$
$$+ \sum_{a=1}^A \sum_\tau \sum_b \sum_c \sum_d e_{b,c,d}^{a,\tau} \Psi_{a,b,c,d}^\tau(j, k, l) \tag{3}$$

where A is the count of the decomposition levels, $\Phi_{a,b,c,d}^0(x, y, z)$ is the scaling function, $h_{b,c,d}^A$ is scaling co-efficients, and $e_{b,c,d}^{a,\tau}$ represents the high-frequency details at different scales/directions. The seven orientation groups of detail coefficients and one group of approximation coefficients is the output of the decomposition technique (Eq. (3)) of the signal $f(j, k, l)$ with size $N \times N \times N$. The signal is downsampled in each direction during the decomposition and the size of the resulted collection of coefficients is $N/2 \times N/2 \times N/2$.

### Shearlets ($H_T$)

In multivariate problems, shearlets are obtained from multi-resolutional analysis and progressive encoding of image edges/ridges. In 2010, the discrete shearlets were expanded (*Xu, Ai & Wu, 2013*) from the Parseval shearlet frames by adjoining additional basis functional components to get orthonormal functions for each shear-matrix. The shearlet transform for $f \in L(\Re^3)$ is given by:

$$\langle f, \phi_{j,\ell,k}^{(d)} \rangle = \int_{\Re^3} \hat{f}(\xi)(W^{-2j}\xi) U_{j,\ell}^{(d)}(\xi) e^{2\pi i \xi A_{(d)}^{-j} B_{(d)}^{[-\ell]} k} d\xi \tag{4}$$

where $j \geq 0$, $\ell = (\ell_1, \ell_2)$ with $|\ell_1|, |\ell_2| \leq 2^j$, $k \in \mathscr{Z}^3$, $\hat{f}(\xi)$ is the fourier transform of the signal $f$ with variable $\xi$, $W^{-2j}\xi$ is a scaling term of the frequency $\xi$ ($j$ controls the scale), $U_{j,\ell}^{(d)}$ represents the pyramidal regions and directional index (d = 1, 2, 3.3), $e^{2\pi i \xi A_{(d)}^{-j} B_{(d)}^{[-\ell]} k}$

represents the oscillatory component of $f$ essential for localizing the image's features in space/frequency, $A$ and $B$ are scaling and shear matrices.

### Curvelets ($C_T$)

Curvelets are also extended from wavelets and are developed to manipulate minute curves with smaller coefficients. As the wavelets are indexed by position and scale, the curvelet-frames use different locality, degrees, and scales. This enhances the performance of curvelets in data-signal classification. Multiscale ridgelet functions with varied-scale bandpass filtering are utilized to generate curvelets (*Candes et al., 2006*). They are evaluated at all locations, angles, and scales and obey parabolic-scale relationship (for solving small structures like edges and ridges in an image). Discrete curvelet transform $C^D(a, b, c)$ at scale a, position b, and orientation c is given by:

$$C^D(a, b, c) = \sum_{n_1, n_2} f(n_1, n_2) \varnothing_{a,b,c}^D(n_1, n_2) \tag{5}$$

where $f$ is the image; $f(n_1, n_2), 0 \leq n_1, n_2 < n$ is in the cartesian grid, and $\varnothing_{a,b,c}^D$ is the curvelet basis function.

### Scattering transform ($S_T$)

The scattering transform (*Adel et al., 2017*) can be explained by a sequence of multiscale/multidirectional gabor-wavelet transforms that are positioned between layers with modulus nonlinearities. The result of these layers mimics a no-parameter convolutional neural network and works well in sparse/low data environment. The computation process of this transform can be represented as the multi-path signal scattering with several wavelet basis functions in a deep convolutional network. In the first layer 0, the Gaussian blur filter $\phi_K(x)$ is applied to $x$. The first scattering representation contains these means of the subsampled coefficients. The calculated coefficients $Sf$ of scattering transform at each layer is given by:

$$\begin{aligned} D_f = \{f * \phi_j, \{|f * \psi_{\lambda_1}| * \phi_j\}\lambda_1, \\ \{||f * \psi_{\lambda_1}| * \psi_{\lambda_2}| * \phi_j\}_{\lambda_1, \lambda_2}\} \end{aligned} \tag{6}$$

where $f * \phi_j$ denotes the signal $f$ convolved (*) with a low-pass filter $\phi_j$, $f * \psi_{\lambda_1}$ is the signal $f$ convolved with a band-pass filter $\psi_{\lambda_1}, \psi_{\lambda_1}, \psi_{\lambda_2}$ are the wavelet functions that act as band-pass filters, and $||f * \psi_{\lambda_1}| * \psi_{\lambda_2}|$ is the higher-order scattering term that captures interactions between different frequency components.

## NORMALIZED GROUP ACTIVATIONS BASED FEATURE EXTRACTION

In this study, our main aim is to develop a novel group activations based interpretable feature for AD detection. The proposed process is split up into two networks: Multi-trait condensed feature extraction networks and normalized group activations for regional association networks. Initially, multiple condensed features are obtained from the first

component by processing the MRI or fMRI time-series data of individual brain regions that are mapped to the AAL template. Subsequently, these features are used to create several regional association networks using normalized group activations. It outputs the correspondence between two individual regions of the brain. At last, these features are used for classification.

This technique can be used for both structural MRI and fMRI time-series. This method is inspired by FCNet (*Riaz et al., 2020*) which is an end-to-end model that inputs the fMRI time-series data and creates features using a regional connectivity/similarity network to identify Attention Deficit Hyperactivity Disorder (ADHD) subjects from normal controls. But they have used only fully connected layers for feature extraction and classification. Here, their model is improved by adding the multi-trait condensed feature extraction as the initial process and a transformer at the end. The AD features are explained/interpreted using functional connectivity connectograms and class-specific transformer visualizations.

Our model is different from FCNet in four ways: First, FCNet only uses rs-fMRI data, while our model uses additional information from both structural MRI and functional rs-fMRI. The multimodal approach captures brain structural and functional features effectively, meaning that more neurodegenerative traits are generally vital in the progression of diseases such as Alzheimer's. The end-to-end integration of richer features is thus produced than with FCNet's focus on rs-fMRI alone. Second, FCNet bases connectivity between brain regions from time-series data from rs-fMRI on CNNs. On the other hand, our model extracts varied features from MRI and fMRI, which elaborates in depth the structural and functional aspects of brain pathology. Third, our approach differs from FCNet because we deal with activations in groups, thus embedding complex interactions between the different regions involved. Network-level modeling like this is perfectly aligned with the nature of Alzheimer's pathology, where multiple regions degenerate simultaneously. Fourth, FCNet is lacking in interpretability as it ranks only functional connections. Our model provides class-specific attention and visualizations that lead to understanding which regions of the brain are driving the diagnosis in such cases, thereby providing better explainability in clinical settings.

## Multi-trait condensed features extraction networks

Multi-trait condensed features extraction networks represent the multi-layered CNNs from different deep learning models that output the features from MRI or fMRI time-series data of individual brain regions. The flow is depicted in Fig. 1. This component's input varies based on the number of points in the individual regions.

- For fMRI data, let us consider there are $np_1$, $np_2$, ..., $np_k$ points in regions $R_1$, $R_2$, ..., $R_k$ with each point containing $nr_t$ time-series data. For every $R_i$, an array of size $np_i \times nr_t$ is created and resized to $P \times P \times 3$. For example, the $nr_1$ of region $R_1$ is 1,000 and $nr_t$ is 140, then an array of size $220 \times 220 \times 3$ is created and used for the next stage.
- For MRI data, let us consider the regions $R_1$, $R_2$, ..., $R_k$ contains $np_1$, $np_2$, ..., $np_k$ points. For every $R_i$, an array of size $np_i$ is created and resized to $Q \times Q \times 1$. Suppose the

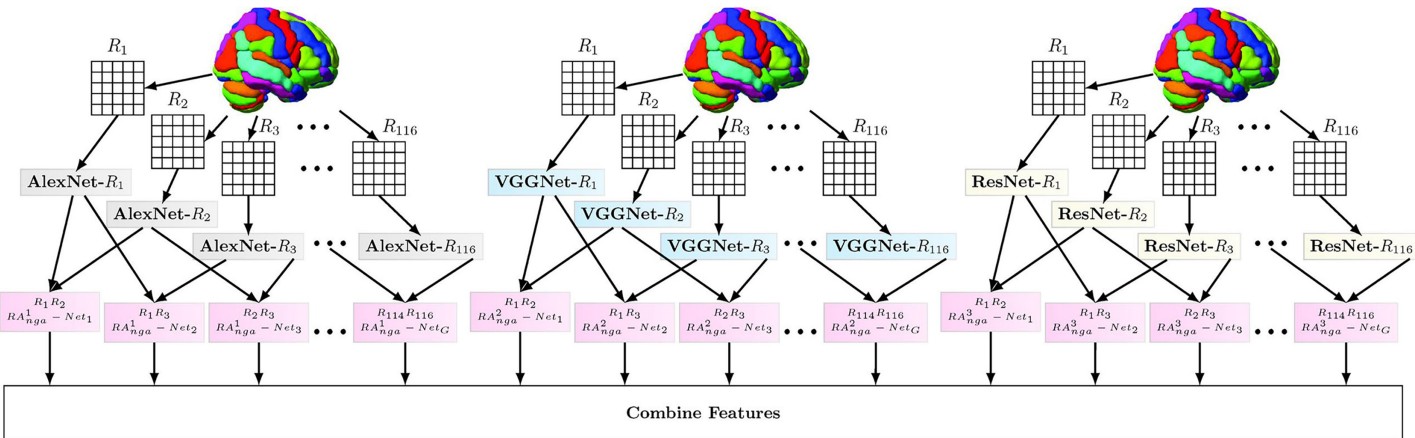

**Figure 1  Multi-Trait Condensed Features Extraction Technique with Normalized Group Activation based Regional Association Estimation.**
$RA^1nga$, $RA^2nga$, $RA^3nga$ indicates the region pair association networks trained from features of Alex Net, VGG Net, and Res Net, respectively.

region $np_1$ of $R_1$ is 1,500, then an array of size $40 \times 40 \times 1$ is created and used for the next stage of processing. Based on the input size, the number of maxpooling layers with stride 2 is altered accordingly.

For this study, the AlexNet convolutional network (Alex_Net) (*Chen et al., 2021*), Visual Geometry Group networks (VGG_Net) (*Chen et al., 2021*), and Residual neural networks (Res_Net) (*Chen et al., 2021*) are used for extracting features for the next stage. The final layers of these networks are slightly modified to suit feature extraction purposes. In Alex_Net, The final dense and softmax blocks are replaced by a single dense block with 256 units. The trainable parameters for this configuration is 3,789,760. In VGG_Net-16, the final three dense and softmax blocks are replaced by a single dense block with 256 units. The total training parameters of VGG_Net-16 feature extraction module is 3,789,760. Res_Net-18 is used and the parameters to be trained is 11,304,576. The final dense, and softmax blocks are replaced by a single dense block with 256 units.

## Regional association estimation networks with normalized group activations

Here, the main idea is to allow the network to use activations from a collection of functions for each neuron in its every layer. For example, in a convolution block, there are several activations that are chosen, applied, normalized and weights are assigned to each function. These types of activations (*Klabjan & Harmon, 2019*) are successfully implemented in MNIST, ISOLET, and CIFAR-100 datasets. This study explores the activational features in the field of neuroimaging for AD classification using multiple cohorts. The following process details the steps involved in creating such layers with normalized activations. Here, the main problem is the unbounded nature of functions like inverse absolute value, and rectifiers. However, a simple addition will lead to error. This issue can be solved by the

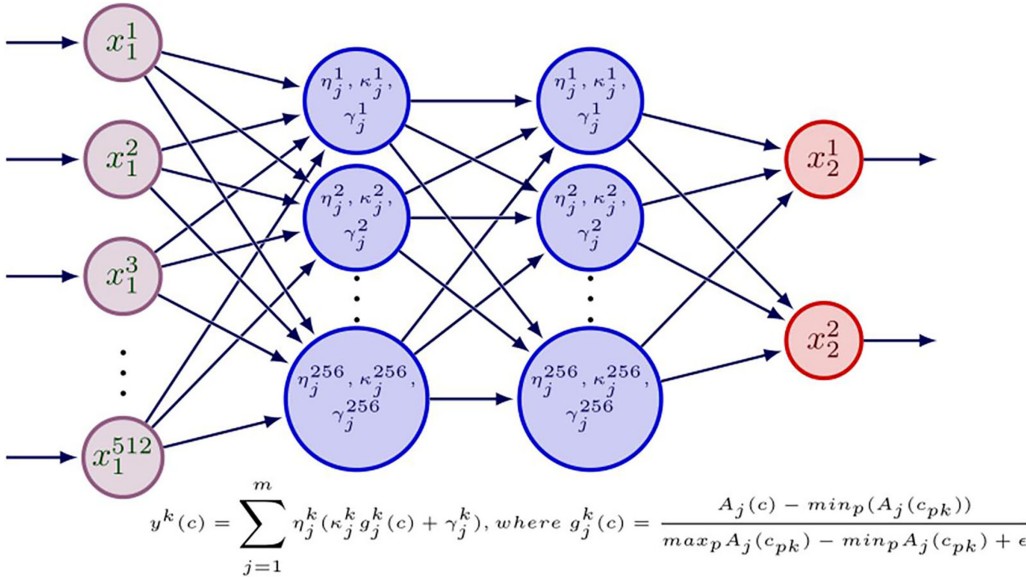

$$y^k(c) = \sum_{j=1}^{m} \eta_j^k (\kappa_j^k g_j^k(c) + \gamma_j^k), \; where \; g_j^k(c) = \frac{A_j(c) - min_p(A_j(c_{pk}))}{max_p A_j(c_{pk}) - min_p A_j(c_{pk}) + \epsilon}$$

**Figure 2 Configuration of a normalized group activations based regional association network ($RA_{nga}$ – $Net$) between two regions.** $x_1$ and $x_2$ are the inputs and outputs. $\eta$, $\kappa$, $\gamma$ are the variables to be trained. $A$ is the set of $m$ activations used and $g$ is normalization function.

normalization of various activations to provide an equivalent proportion in training. The output of each neuron $k$ with $m$ activation functions ($a_m \; \varepsilon \; A$) is given by:

$$
\begin{aligned}
y^k(c) &= \sum_{j=1}^{m} \eta_j^k g_j^k(c), \; where \; g_j^k c \\
&= \frac{A_j(c) - min_v(A_j(c_{vk}))}{max_v A_j(c_{vk}) - min_v A_j(c_{vk}) + \varepsilon}
\end{aligned}
\tag{7}
$$

In the Eq. (7), $p$, $c_{vk}$, and $\eta_j^k$ denotes the number of samples in a mini-batch, the $i^{th}$ neuron input with $v^{th}$ sample, $g_j^k(c)$ is the normalized value of the $j^{th}$ activation function for neuron $k$, $A_j(c)$ is the output of the $j^{th}$ activation function for the input sample $c$, and $max_v(A_j(c_{vk}))$, $min_v(A_j(c_{vk}))$ are maximum/minimum values of the activation function across the mini-batch. To avoid any disproportionate increase in the magnitude of a particular activation, both the weights $\eta$ and all the activation values have to be restricted between $[0, 1]$. Here, we use the optimization of $\eta_j$ and the loss calculation based on *Klabjan & Harmon (2019)*. $\kappa$ and $\gamma$ are the additional parameters that are used to decide on the original state of the activation.

$$y^k = \sum_{j=1}^{m} \eta_j^k (\kappa_j g_j^k + \gamma_j). \tag{8}$$

To summarize, the normalized group activations are computed using a two-stage process. First, the min-max values of the activations need to be obtained using a

mini-batch. Second, the parameters $\eta$, $\kappa$, and $\gamma$ are learned during the training using full data using the Eq. (8). These parameters are calculated only on the train dataset.

Various Siamese similarity networks (*Bromley et al., 1994*) with the normalized group activations that estimate the correspondence between the extracted features for a pair of brain regions. The configuration of individual network is given in Fig. 2. These are called regional association estimation networks. This can be taken as the strength of the association/connection between the regional pair. Each network processes the features of two regions extracted from multi-trait regional condensed features extraction networks ("Multi-trait Condensed Features Extraction Networks"). Here, features from each pair of 116 AAL regions are given as input to $RA_{nga} - Nets$. A total of $G = 6,670\ RA^i_{nga} - Nets$ for all the combinations of all the pairs of 116 AAL regions are created. Their output is given as the input for the next stage classification process.

## AUTOMATED AD DIAGNOSIS SYSTEM USING HETEROGENEOUS FEATURES ($T_f^{NGA-RA}$)

We expand our research by evaluating the performance of the proposed features on heterogeneous data. For this purpose, an automated AD diagnosis system is designed to extract/test the normalized group activations features on seven heterogeneous features (Non-Transformed ($N_T$), 3D-Wavelets ($W_T$), RROI, GLCM, Curvelets ($C_T$), Shearlets ($H_T$), Scattering Transform ($S_T$)). Figure 3 shows the overview of the whole process and is represented as $T_f^{NGA-RA}$.

The initial features are obtained from Alex_Net, VGG_Net, and Res_Net (refer to "Normalized Group Activations Based Feature Extraction") and are denoted as $RA^1_{nga}$, $RA^2_{nga}$, $RA^3_{nga}$. The outputs are combined and then resized into $Tf_n \times Tf_n \times 3$ arrays. It is split into patches and used as input for the vanilla vision transformer (*Dosovitskiy et al., 2021*). The patches are then linearly projected/positioned embedded and fed into multi-head self-attentions blocks. Subsequently, it is given into the MLP (multi-layer perceptron) layer to minimize/optimize the cross entropy loss and predict the multi-stage AD. As a pre-training step, calculation of minimum/maximum values for the activations used is done on a mini-batch for each classification process.

The activations analyzed in this study are Linear Rectifier ($max(0, x)$), Exponential Linear ($x$ for $x \geq 0$ and $e^x - 1$ for $x \leq 0$), and ELiSH (Exponential Linear Sigmoid SquasHing) (*Basirat & Roth, 2018*) $\left(x = \frac{e^x - 1}{1 + e^x}\ if\ x \geq 0\right)$. It involves the creation of a set of commonly used functions.

## EXPERIMENTAL SETUP AND RESULTS

The proposed model ($T_f^{NGA-RA}$) is implemented/executed using Python Tensorflow library and the system consists of two RTX 3080 NVIDIA GPUs and Intel i7 processor. Open-source libraries are used for heterogeneous data transformations. Initially, all the images in the dataset are preprocessed (refer to "New-Texture-Extraction Model (RROI)" & "GLCM Textures (GLCM)") and mapped to AAL template. For the pre-training phase (calculation of minimum/maximum values for the $m = 3$ activations): 20% of the training set is used as a mini-batch (equal number of subjects for all classes in the classification). Pre-training

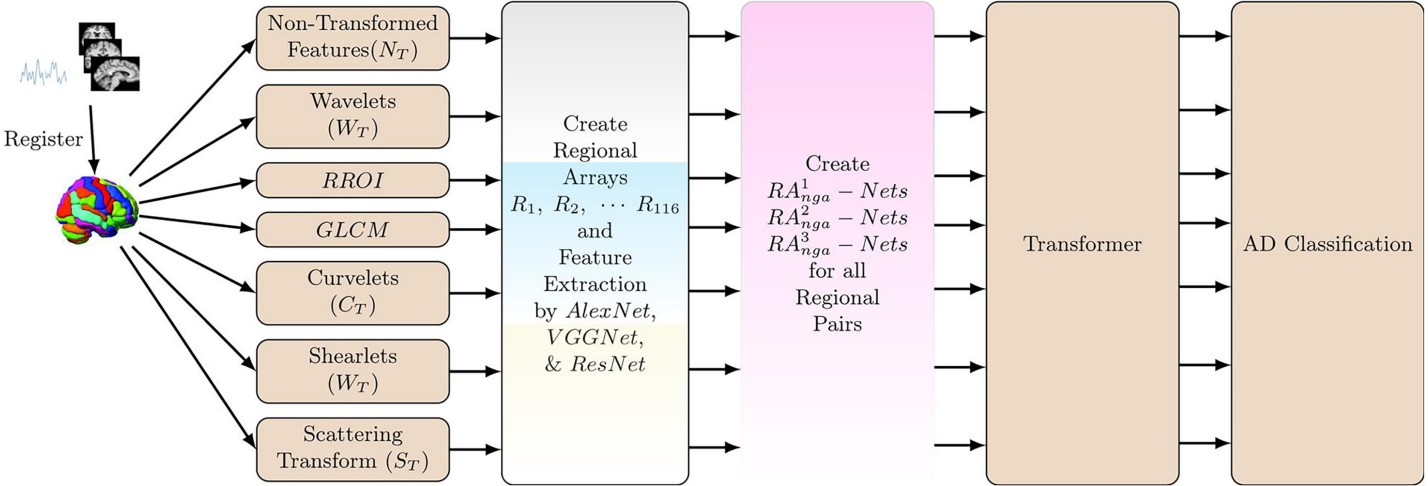

**Figure 3  Block Diagram of Automated AD Diagnosis System using Heterogeneous Features ($Tf^{NGA-RA}$).** Let $N_T$, $W_T$, RROI, GLCM, $C_T$, $H_T$, $S_T$ denotes the seven heterogeneous features such as Non-Transformed features, 3D-Wavelets, RROI, GLCM, Curvelets, Shearlets, and Scattering Transform, respectively.

Settings: optimizer: Adam, learning rate = 1e−4 $\beta_1$ = 0.9, $\beta_2$ = 0.99, batch size = 5, and number of epochs = 15. The initial weights of two layers are set to the value $\frac{1}{m}$ with $\kappa = 0$ and *gamma* = 1. The initial value of the $\alpha$ variable is based on He Normal method. For the main training phase: the Transformer is implemented with settings: optimizer: Adam, learning rate = 1e−5 $\beta_1$ = 0.9, $\beta_2$ = 0.99, batch size = 5, and number of epochs = 50.

For each feature, the trained model is evaluated on a new test set. This process measures both the efficiency of the model and the corresponding feature involved in training the network. For binary classifications, the results metrics are presented as follows:

$$Specificity = \frac{TN}{TN + FP} \tag{9}$$

$$Sensitivity = \frac{TP}{TP + FN} \tag{10}$$

$$Accuracy = \frac{TP + TN}{TP + TN + FP + FN} \tag{11}$$

where TP, FP, TN, and FN represent the true-positives, false-positives, true-negatives, and false-negatives, respectively.

For ternary classification, the metrics indicating the precision, recall and $F_1$-score values of individual classes are:

$$Precision_1 = \frac{T_1}{T_1 + T_{12} + T_{13}} \tag{12}$$

$$Recall_1 = \frac{T_1}{T_1 + F_{12} + F_{13}} \tag{13}$$

$$F_1 - score = 2 \cdot \frac{Precision_1 \cdot Recall_1}{Precision_1 + Recall_1} \tag{14}$$

**Table 2 Summarizes the best performance of proposed model ($Tf^{NGA-RA}$) in different classifications using rs-fMRI.**

| Type | Feat. | Acc | Sens | Spec | AUC | |
|------|-------|-----|------|------|-----|---|
| AD *vs.* NC | $N_T$ | 0.966 | 0.942 | 0.959 | 0.972 | |
| | $W_T$ | 0.953 | 0.935 | 0.945 | 0.961 | |
| | $S_T$ | 0.950 | 0.949 | 0.921 | 0.958 | |
| $e_{MCI}$ *vs.* | $W_T$ | 0.817 | 0.836 | 0.801 | 0.837 | |
| $l_{MCI}$ | $N_T$ | 0.799 | 0.821 | 0.800 | 0.815 | |
| AD *vs.* $e_{MCI}$ | $N_T$ | 0.839 | 0.851 | 0.820 | 0.846 | |
| AD *vs.* $l_{MCI}$ | $W_T$ | 0.823 | 0.847 | 0.816 | 0.828 | |
| NC *vs.* $e_{MCI}$ | $N_T$ | 0.828 | 0.842 | 0.815 | 0.833 | |
| | $W_T$ | 0.819 | 0.819 | 0.830 | 0.828 | |
| NC *vs.* $l_{MCI}$ | $N_T$ | 0.820 | 0.815 | 0.831 | 0.825 | |
| | $W_T$ | 0.802 | 0.810 | 0.799 | 0.810 | |
| | | Acc | Class | F1 | Prec | Rec |
| AD *vs.* NC | $N_T$ | 0.811 | AD | 0.842 | 0.780 | 0.938 |
| *vs.* MCI | | | NC | 0.824 | 0.770 | 0.791 |
| | | | MCI | 0.811 | 0.759 | 0.777 |

$$Accuracy = \frac{T_1 + T_2 + T_3}{T_1 + T_2 + T_3 + F_{12} + F_{13} + F_{21} + F_{23} + F_{31} + F_{32}} \qquad (15)$$

where $T_1, T_2, T_3$ are the true positives for classes 1, 2, and 3, respectively, $F_{ij}$ are the false negatives which are instance of class i that were incorrectly classified as class j.

rs-fMRI performance on heterogeneous features is evaluated by randomly selecting 70% of the minority class subjects from the whole fMRI set. Then, the same number of the majority class subjects is also randomly selected and combined to form the training set. A total of 10% of random images are used for validation and the remaining is used as the test set. Classifications done are AD *vs.* NC, AD *vs.* $e_{MCI}$, AD *vs.* $l_{MCI}$, NC *vs.* $e_{MCI}$, NC *vs.* $l_{MCI}$, and AD *vs.* NC *vs.* MCI. For the MRI dataset, the evaluation process on heterogeneous features is given by selecting 90% of the minority class subjects from the whole ADNI-1 set. Then, the same number of the majority class subjects is also randomly selected and combined to form the training set. The remaining images are used for validation. The full ADNI-2 set is used as the test set. The above mentioned process is also repeated for ADNI-2 as training/validation. The full ADNI-1 set is used as the test set. Classifications done are AD *vs.* NC, AD *vs.* cMCI, AD *vs.* MCInc, NC *vs.* cMCI, NC *vs.* MCInc, and AD *vs.* NC *vs.* MCI.

For both fMRI and MRI datasets, to measure the generality of the heterogeneous features and the deep learning model, the evaluation process specified above is performed on 25 randomly generated sets and the mean result is reported. The results of the binary classification is represented as (accuracy, sensitivity, specificty). For example, the result [0.92, 0.90, 0.94] indicates 0.92 of accuracy, 0.90 of sensitivity, and 0.94 of specificity. For

ternary classification, the results are shown as (Accuracy, F1-score of AD, F1-score of NC, F1-score of MCI).

## Classification results-rs-fMRI

Table 2 highlights the best performance in different classifications using fMRI. For AD *vs.* NC, $N_T$ gives the best performance of [0.966, 0.942, 0.959]. $W_T$ and $S_T$ are almost equal to $N_T$ with [0.953, 0.935, 0.945] and [0.950, 0.949, 0.921]. The min/max accuracy of $N_T$ and $W_T$ is around 0.950/0.978. For differentiating between $e_{MCI}$ and $l_{MCI}$, the best is given by $W_T$ with [0.817, 0.836, 0.801] and $N_T$ is the closest one with $W_T$ reaching [0.799, 0.821, 0.800].

For AD *vs.* $e_{MCI}$, $N_T$ gives the best performance of [0.839, 0.851, 0.820] with a 1–7% increase in performance over other features. $W_T$ and $C_T$ stands second with [0.830, 0.810, 0.841] and [0.811, 0.833, 0.796]. For AD *vs.* $l_{MCI}$, $W_T$ and $N_T$ performs better (1–10%) than other features with [0.823, 0.847, 0.816] and [0.813, 0.806, 0.853].

For NC *vs.* $e_{MCI}$, $N_T$ sores highest with [0.828, 0.842, 0.815] and indicates a 1–17% increase in accuracy over other features. $W_T$ and $C_T$ yield second best with [0.819, 0.815, 0.830] and [0.780, 0.781, 0.806]. For NC *vs.* $l_{MCI}$, $N_T$ achieves the highest accuracy with [0.820, 0.815, 0.831] (2–17% increase in accuracy than other features) and $W_T$ reaches [0.802, 0.810, 0.799]. For AD *vs.* NC *vs.* MCI, $N_T$ trained model reaches the highest [0.811, 0.842, 0.824, 0.811] (1–17% increase in accuracy than other features).

It is inferred that the $N_T$, $W_T$ and $C_T$ produce better accuracy than the other features. $S_T$ shows good performance only for AD *vs.* NC and performs better than $H_T$. In texture-oriented features (*RROI* and *GLCM*), *GLCM* shows better accuracy by 1–3%. $H_T$ gives the lowest performance.

## Classification results-MRI: train set: ADNI-1, test set: ADNI-2

Table 3 shows the performance of $Tf^{NGA-RA}$ in different classifications using ADNI-1 as training data. For AD *vs.* NC, $N_T$ beats all the other features with [0.940, 0.941, 0.932]. $W_T$ and *GLCM* reaches [0.919, 0.923, 0.919] and [0.902, 0.891, 0.883]. *RROI*, $C_T$, and $S_T$ yield [0.902, 0.889, 0.903], [0.907, 0.881, 0.901], and [0.889, 0.871, 0.903]. For NC *vs.* cMCI, the model trained with $N_T$ gives the highest performance of [0.784, 0.781, 0.750] which is a 3–17% increase in performance. The next good result is obtained by $W_T$ with [0.751, 0.761, 0.766]. Distinguishing NC from MCInc, $W_T$ beats the other features by a 1–13% increase in accuracy with [0.753, 0.740, 0.761]. $N_T$ gives the closest result of [0.742, 0.738, 0.749]. In the AD and cMCI category, Both $N_T$ and $W_T$ yield good results with [0.739, 0.765, 0.720] and [0.721, 0.745, 0.711] which is a 3–15% increase in performance than other features. $H_T$ and $S_T$ give [0.585, 0.563, 0.587] and [0.576, 0.569, 0.543] which is the lowest in this category. For AD *vs.* MCInc, $N_T$, $W_T$ beats all the other features by 1–13% with the score of [0.731, 0.722, 0.764] and [0.741, 0.749, 0.741]. For AD *vs.* NC *vs.* MCI, $N_T$ trained model outperforms the other features by 3–11% with [0.771, 0.802, 0.810, 0.751]. $W_T$ obtains the second maximum of [0.746, 0.781, 0.783, 0.743] and $C_T$ reaches [0.735, 0.751, 0.669, 0.727].

Table 3 Summarizes the best performance of proposed model ($Tf^{NGA-RA}$) in different classifications using ADNI-1 as training data.

| Type | Feat. | Acc | Sens | Spec | AUC |
|---|---|---|---|---|---|
| AD *vs*. NC | $N_T$ | 0.940 | 0.941 | 0.932 | 0.956 |
| NC *vs*. cMCI | $N_T$ | 0.784 | 0.781 | 0.750 | 0.804 |
| NC *vs*. | $W_T$ | 0.753 | 0.740 | 0.761 | 0.760 |
| MCInc | $N_T$ | 0.742 | 0.738 | 0.749 | 0.753 |
| AD *vs*. cMCI | $N_T$ | 0.739 | 0.765 | 0.720 | 0.747 |
| | $W_T$ | 0.721 | 0.745 | 0.711 | 0.738 |
| AD *vs*. | $N_T$ | 0.731 | 0.722 | 0.764 | 0.740 |
| MCInc | $W_T$ | 0.741 | 0.749 | 0.740 | 0.746 |

| Type | Feat. | Acc | Class | F1 | Prec | Rec |
|---|---|---|---|---|---|---|
| AD *vs*. NC | $N_T$ | 0.771 | AD | 0.802 | 0.710 | 0.909 |
| *vs*. MCI | | | NC | 0.810 | 0.700 | 0.735 |
| | | | MCI | 0.751 | 0.731 | 0.699 |

Table 4 Summarizes the best performance of proposed model ($Tf^{NGA-RA}$) in different classifications using ADNI-2 as training data.

| Type | Feat. | Acc | Sens | Spec | AUC |
|---|---|---|---|---|---|
| AD *vs*. NC | $N_T$ | 0.900 | 0.875 | 0.931 | 0.937 |
| | $S_T$ | 0.892 | 0.875 | 0.900 | 0.912 |
| NC *vs*. | $N_T$ | 0.710 | 0.691 | 0.749 | 0.720 |
| MCInc | $C_T$ | 0.701 | 0.671 | 0.759 | 0.719 |
| AD *vs*. | $W_T$ | 0.783 | 0.742 | 0.820 | 0.780 |
| MCInc | $N_T$ | 0.769 | 0.739 | 0.789 | 0.788 |

| Type | Feat. | Acc | Class | F1 | Prec | Rec |
|---|---|---|---|---|---|---|
| AD *vs*. NC | $N_T$ | 0.727 | AD | 0.710 | 0.629 | 0.830 |
| *vs*. MCI | | | NC | 0.699 | 0.659 | 0.741 |
| | | | MCI | 0.743 | 0.839 | 0.667 |
| | $W_T$ | 0.715 | AD | 0.681 | 0.620 | 0.766 |
| | | | NC | 0.694 | 0.651 | 0.730 |
| | | | MCI | 0.737 | 0.817 | 0.664 |

It is inferred that the $N_T$ and $W_T$ consistently outperform all the other features. The textures features *RROI* and *GLCM* perform better than $C_T$, $H_T$ and $S_T$ in most classifications.

## Classification results-MRI: train set: ADNI-2, test set: ADNI-1

As there are only 38 baseline cMCI are selected for ADNI-2, the considered classifications are AD *vs*. NC, NC *vs*. MCInc, AD *vs*. MCInc, and AD *vs*. NC *vs*. MCI. Table 4 illustrates the peak performance $Tf^{NGA-RA}$ in different classifications using ADNI-2 as training data. In the AD *vs*. NC category, $N_T$ and $S_T$ models reach the highest of [0.900, 0.875, 0.931] and

[0.892, 0.875, 0.900]. $W_T$ and $C_T$ stand next with [0.875, 0.915, 0.864] and [0.870, 0.906, 0.859]. For NC *vs.* MCInc, $N_T$ and $C_T$ exceed the performance of other features by 2–9% with a score of [0.710, 0.691, 0.749] and [0.700, 0.671, 0.759]. $W_T$ reaches [0.683, 0.656, 0.723].

For AD *vs.* MCInc, better classification results of [0.783, 0.742, 0.820] with $W_T$, [0.769, 0.739, 0.789] with $N_T$. They perform better than others by 1–12%. For the ternary classification (AD *vs.* NC *vs.* MCI), the performance of [0.727, 0.710, 0.699, 0.743] and [0.715, 0.681, 0.694, 0.737] is given by $N_T$ and $W_T$. They beat other features by 3–14%. The inferences from the results are: $N_T$, $W_T$, and $C_T$ outperforms the other features for all the classifications.

# DISCUSSION

In this study, there are different types of features analyzed in the domain of AD classification. From the results, it is shown that non-transformed and wavelet features outperform all the other transformations. Nevertheless, in this task, additional specialised transformations like textures, shearlet, curvelet, and scattering have not always attained similar accuracy. This disparity is attributed to other variables. Even while they are excellent at catching particular kinds of characteristics or structures in signals or images, but did not match up well with the patterns linked to AD pathology. On the other hand, wavelet provides a more flexible method that is able to capture a wider variety of characteristics, which makes it more appropriate for capturing the intricate fluctuations found in AD imaging. Furthermore, wavelet characteristics is more resilient to noise and unpredictability in imaging data than texture, shearlet, curvelet, and scattering transformations. These specialised transformations require extra processing steps that are not in scope of this study.

The interpretability of the retrieved characteristics is another crucial factor to take into account. Understanding the mechanisms connected to AD, as well as determining pertinent features for classification, depend on interpretability. In contrast to wavelet, shearlet, curvelet, and scattering transformations, non-transformed characteristics offer a more comprehensible depiction of the underlying data. To summarise, non-transformed features, which strike a compromise between efficacy and computing efficiency, frequently give a more dependable and useful solution for AD classification tasks.

In this section, we analyze the visualization/performance of the proposed feature extraction technique and $Tf^{NGA-RA}$ model using rs-fMRI and MRI features. For each epoch, the training consists of two phases: first, the feature generation from multiple regional association networks; second, the main transformer training phase. Their deep learning configuration is explained in "Experimental Setup and Results". On the whole, this combined training process for each epoch is 29 min and the total time taken to complete 50 epochs is approximately 24 h. The prediction of an unknown image takes around 28 s.

For evaluation purpose, $Tf^{ReLU-RA}$ transformer model is created. It replaces the normalized group activations in the layers with only ReLU units and all the other configurations are the same as $Tf^{NGA-RA}$. The study (*Klabjan & Harmon, 2019*) shows

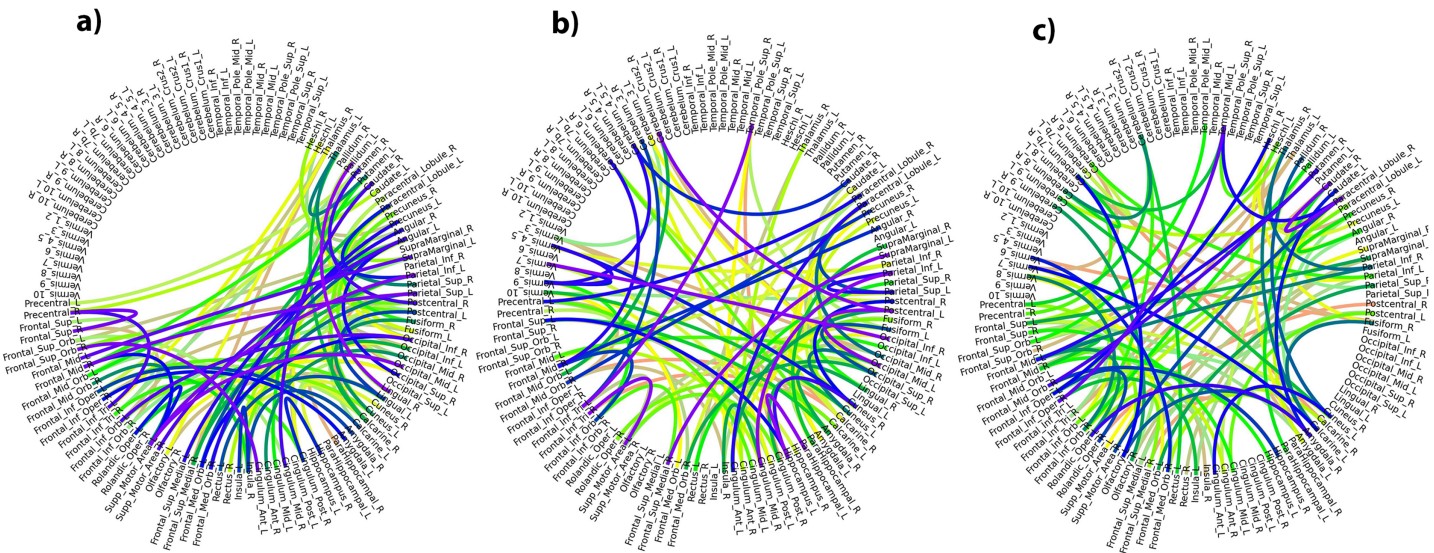

**Figure 4** (A–C) Depicts regional feature associations ($N_T$) produced by $RA_{nga}$–*Nets* for AD *vs* NC, $e_{MCI}$ *vs* $l_{MCI}$, and *NC vs* $e_{MCI}$, respectively.

a set of activations can improve results in various datasets. It is understood that this is problem-specific. Here, using the $Tf^{NGA-RA}$ model, we want to evaluate the normalized group activations features in multi-stage AD classifications using rs-fMRI/MRI data.

## rs-fMRI analysis

For rs-fMRI, the output of $RA_{nga} - Nets$ can be considered as the connection strength of the regional-pairs (inspired by Siamese similarity networks (*Bromley et al., 1994*)). For each random run, there are slight variations in features selected, so the common regional pairs with high connection power found in all the runs are taken into consideration. At first, the features generated by $RA_{nga} - Nets$ is investigated for AD-specific disease classification. For three classifications (AD *vs.* NC, $e_{MCI}$ *vs.* $l_{MCI}$, and NC *vs.* $e_{MCI}$) using $N_T$, the top-100 connection strengths obtained by $RA_{nga} - Nets$ are depicted on an AAL template in Fig. 4. The color of connections denotes the connectivity strength between any two regional nodes. The dark color indicates high strength.

For AD *vs.* NC, the top connections of $RA_{nga} - Nets$ ($N_T$) are associated with regional pairs of frontal-superior, frontal-middle, para-hippocampal, and frontal-superior-medial. For MCI classifications, it is noted that the connections related to hippocampal, middle-temporal gyral, superior-frontal gyral, precuneus, medial-orbitofrontal gyral, para-hippocampal, and supra-marginal gyral regional pairs are found to have high connectivity values. It is worth mentioning that other AD-specific research (*Zhang et al., 2021*) also conform that these regions show good discriminative power in MCI classifications which further authenticates $RA_{nga} - Nets$ ($N_T$) connections. These regional pairs are shown in Table 5 which are conformed in previous studies.

Table 5 **Comparison of recent methods with our model ($Tf^{NGA-RA}$) using ADNI rs-fMRI database.**

| Year & study | | | | | Features |
|---|---|---|---|---|---|
| *Khatri & Kwon (2022a)* | #63AD, #68NC, 0.958 | #68NC, #82MCI, 0.924 | #63AD, #82MCI, 0.903 | | Structural/Functional MRIbiomarkers |
| *Ghanbari et al. (2023)* | #49AD, #49NC, 0.937 | #49NC, #49MCI, 0.968 | #49AD, #49MCI, 0.917 | | Node Redundancy metrics |
| *Chen et al. (2024)* | | | | #25MCI-C, #77MCI-NC, 0.86 | Multi-modal MRI characteristics |
| *Hojjati et al. (2017)* | | | | #18MCI-C, #62MCI-NC, 0.914 | Connectivity matrices, local/global graph-measures |
| *Dong et al. (2021)* | | | | #48$e_{MCI}$, #50$l_{MCI}$, 0.784 | Hybridfeatures |
| $Tf^{ReLU-RA}$ | #34AD, #52NC, 0.971 | #52NC, #59$e_{MCI}$, 0.804 #52NC, #40$l_{MCI}$, 0.810 | #34AD, #40$l_{MCI}$, 0.799 | #59$e_{MCI}$, #40$l_{MCI}$, 0.791 | |
| Proposed $Tf^{NGA-RA}$ | #34AD, #52NC, 0.966 | #52NC, #59$e_{MCI}$, 0.828 #52NC, #40$l_{MCI}$, 0.820 | #34AD, #40$l_{MCI}$, 0.823 | #59$e_{MCI}$, #40$l_{MCI}$, 0.817 | |

**Note:**
For a task C1 *vs.* C2, each entry denotes the number of subjects of C1, C2, and the accuracy of the task (C1 *vs.* C2) in that study respectively. MCI-C and MCI-NC indicates MCI subjects that are converted to AD in 6 months and MCI subjects that are not converted to AD in 6 months respectively.

Table 6 **Top cortical regional pairs for AD *vs.* NC.**

| Regional pairs |
|---|
| Hippocampal, Cingulum (*Khatri & Kwon, 2022b*; *Zhang et al., 2021*) |
| Amygdala, Frontal (*Sun, Wang & He, 2022*; *Sheng et al., 2019*) |
| Precuneus, Hippocampal (*Sun, Wang & He, 2022*) |

Table 7 **Top cortical regional pairs for NC *vs.* $e_{MCI}$.**

| Regional pairs |
|---|
| Superior frontal gyrus, Right occipital gyrus (*Bi et al., 2021*) |
| Fusiform gyrus, Left Cuneus (*Sheng et al., 2019*; *Zhang et al., 2019*) |
| Parahippocampal, Insula (*Yang et al., 2019*; *Chen et al., 2016*) |

Table 8 **Top cortical regional pairs for NC *vs.* $l_{MCI}$.**

| Regional pairs |
|---|
| Middle occipital gyrus, frontal gyru (*Sheng et al., 2019*; *Zhang et al., 2019*) |
| Amygdala, Parahippocampal (*Chen et al., 2016*) |
| Precuneus, Occipital gyrus (*Zhang et al., 2019*) |

The results are in Table 2 indicate the high-performing features are $N_T$, $W_T$, and $C_T$. For AD *vs*. NC, the mean connections' strengths of all the regional pairs of the top regions indicated above are as follows: $N_T$–$0.9865 \pm 0.81$, $W_T$–$0.9672 \pm 0.73$, and $C_T$–$0.9414 \pm 0.64$. $S_T$ performs well only for this task. It's mean is $0.9576 \pm 0.79$. For MCI prediction, the mean values are $N_T$–$0.8053 \pm 0.82$, $W_T$–$0.7729 \pm 0.77$, and $C_T$–$0.7498 \pm 0.71$. These results are shown in Tables 6 & 7 and are conformed in other studies. It is noted that these higher values do not imply higher classification accuracy. These values are mainly presented to indicate that higher connection values are obtained from $RA_{nga} - Nets$ for the most discriminative regions of the respective classifications which validate our accuracy.

For AD *vs*. NC, the $Tf^{NGA-RA}$ achieves the accuracy, AUC of 0.966, 0.972 and $Tf^{ReLU-RA}$ yields 0.971, 0.980. There is no significant increase in the performance of the networks. For NC *vs*. $l_{MCI}$ and $e_{MCI}$ *vs*. $l_{MCI}$, $Tf^{NGA-RA}$ accuracy increase by 2% to 3% over $Tf^{ReLU-RA}$. In the NC *vs*. $e_{MCI}$ task, the accuracy and AUC of $Tf^{NGA-RA}$ are 0.828 and 0.833, $Tf^{ReLU-RA}$ reaches 0.791 and 0.804. In AD *vs*. $l_{MCI}$ task, $Tf^{NGA-RA}$ accuracy and AUC is 0.823/0.837 and it is 0.799/0.814 for $Tf^{ReLU-RA}$. Table 8 compares the performance of $Tf^{NGA-RA}$ with recent models using baseline rs-fMRIs. The study (*Khatri & Kwon, 2022a*) integrates both structural MRI and rs-fMRI features to aid AD-classification. rs-fMRI brain networks are combined with sMRI hippocampus subfield and amygdala volume. Brain area connectivity and nodal characteristics are computed using graph theory. Feature selection approaches like random forest and support vector machine classifiers are used to maximise diagnostic accuracy. The article (*Ghanbari et al., 2023*) explores how brain wiring redundancy counteracts aging-related cognitive loss and facilitates early AD detection. It offers a metric to assess redundant disjoint connections in three high-order brain networks using dynamic functional connectivity from rs-fMRI. The results indicate a minor decrease in redundancy from MCI to AD but a considerable increase in redundancy from NC to MCI. An unique framework (*Chen et al., 2024*) that makes use of MRI images' structural and functional brain properties is developed to predict the development from MCI to AD. The alterations in connection, hyperactivity, and atrophy are examined. Multi-modality MRI models provide the highest performance. Deep neural network models are trained from handmade/embedded features (*Dong et al., 2021*) of functional connectivity networks to compare different types of MCI.

The techniques (*Khatri & Kwon, 2022a*; *Chen et al., 2024*) use multi-modal ROI features to improve the accuracies. Also in *Khatri & Kwon (2022a)*, *Ghanbari et al. (2023)*, the collective MCI classifications are handled, whereas in this proposed study, the results of difficult classifications like NC *vs*. $l_{MCI}$, NC *vs*. $e_{MCI}$, AD *vs*. $l_{MCI}$, and $e_{MCI}$ *vs*. $l_{MCI}$ are presented and analysed. The studies (*Hojjati et al., 2017*; *Dong et al., 2021*) use targeted ROIs and hand-picked features, whereas, in this study, multiple regional networks are used to generate features from all regions and specific regions' features are selected automatically. Moreover, with limited subjects, our model shows good accuracy of over 96% for AD *vs*. NC and achieve atleast 81% accuracy in all MCI classifications. Regional association networks improve the understanding of the underlying disease by capturing

a)                                           b)

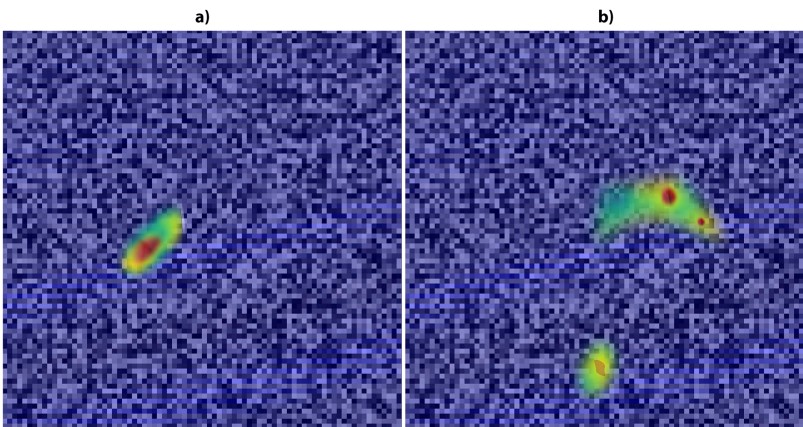

**Figure 5** **(A and B) Depicts the AD-specific and MCI-specific gradient rollout explainability of the transformer.**           

important regional pairs/features (Tables 5–7) and their probabilistic connectivity output enhance the interpretability (Fig. 4) of this proposed model.

To conclude, the features obtained from the proposed normalized group activations feature extraction networks give comparable results with other recent techniques using baseline rs-fMRIs. Also, the results show that the combination of activations can improve the performance of multi-stage AD classifications.

## MRI analysis

For this modality, the features produced by $RA_{nga} - Nets$ can be considered as similarity probabilities of the corresponding regional pairs. After getting the features from $RA_{nga}^1$, $RA_{nga}^2$, $RA_{nga}^3$, The outputs of $RA_{nga}^1$ is resized to $Tf_n \times Tf_n$ array. This process is followed for $RA_{nga}^2$ and $RA_{nga}^3$ features and resized to $Tf_n \times Tf_n \times 3$ array and is used as the input to the transformer module. In this case, we think it is relevant to inspect class specific gradient rollout (*Chefer, Gur & Wolf, 2021*).

Figure 5 depicts the AD-specific and MCI-specific gradient rollout explainability. cMCI/MCInc-specific classifications with respect to AD and NC are averaged to produce MCI-specific gradient rollouts. The red portions denote strong features that are helpful in identifying the AD and MCI classes. From Fig. 5A, the strong AD regional pairs are found to be from the hippocampus and thalamus. Specifically, the important pairs are: (Hippocampus_L, Frontal_Sup_L), (Hippocampus_L, Amygdala_R), (Hippocampus_L, ParaHippocampal_R), (Hippocampus_R, Amygdala_R), (Thalamus_L, ParaHippocampal_R), (Thalamus_L, Hippocampus_L), (Amygdala_L, Hippocampus_R), (Amygdala_R, Thalamus_L). Some weak pairs are: (Temporal_Mid_R, Insula_L), (Temporal_Mid_R, Hippocampus_L), (Temporal_Mid_L, Thalamus_L), (Temporal_Mid_R, Precuneus_R), (Insula_R, Cingulate_Post_L).

The strong MCI regional pairs (Fig. 5B) are found to be from the amygdala, putamen, and parahippocampal areas. The top pairs are: (Amygdala_L, Hippocampus_R), (Amygdala_L, Temporal_Mid_L), (Amygdala_R, Temporal_Mid_L), (Amygdala_L,

**Table 9 Comparison of recent methods with our model ($Tf^{NGA-RA}$) using ADNI-1 and ADNI-2 MRI databases.**

|  | Model | AD *vs.* NC | NC *vs.* cMCI | NC *vs.* MCInc | AD *vs.* cMCI | AD *vs.* MCInc |
|---|---|---|---|---|---|---|
| ADNI-1 | CNN3-D | 0.941, 0.962, 0.914, 0.930 | 0.756, 0.762, 0.731, 0.767 | 0.719, 0.732, 0.699, 0.725 | 0.695, 0.717, 0.684, 0.715 | 0.705, 0.710, 0.694, 0.704 |
|  | VGG3-D | 0.930, 0.951, 0.902, 0.921 | 0.730, 0.742, 0.705, 0.732 | 0.728, 0.743, 0.725, 0.731 | 0.671, 0.701, 0.674, 0.689 | 0.724, 0.736, 0.703, 0.732 |
|  | ResNet3-D | **0.950, 0.975, 0.937, 0.964** | 0.760, 0.765, 0.753, 0.762 | 0.739, 0.753, 0.714, 0.739 | 0.718, 0.725, 0.689, 0.739 | 0.739, 0.765, 0.735, 0.747 |
|  | $Tf^{NGA-RA}$ | 0.945, 0.961, 0.931, 0.959 | 0.766, 0.771, 0.765, 0.750 | 0.739, 0.747, 0.703, 0.725 | **0.746, 0.752, 0.721, 0.745** | **0.750, 0.771, 0.746, 0.757** |
|  | Proposed $Tf^{NGA-RA}$ | 0.940, 0.941, 0.932, 0.956 | **0.784, 0.804, 0.781, 0.750** | **0.753, 0.760, 0.740, 0.761** | 0.739, 0.747, 0.765, 0.720 | 0.765, 0.720, 0.749, 0.740 |
| ADNI-2 | CNN3-D | 0.878, 0.910, 0.882, 0.904 | - | 0.702, 0.679, 0.653, 0.680 | - | 0.748, 0.739, 0.731, 0.750 |
|  | VGG3-D | 0.881, 0.921, 0.892, 0.907 | - | 0.709, 0.694, 0.679, 0.685 | - | 0.758, 0.740, 0.720, 0.759 |
|  | ResNet3-D | **0.925, 0.942, 0.902, 0.933** | - | 0.700, 0.710, 0.681, 0.723 | - | 0.779, 0.755, 0.718, 0.783 |
|  | $Tf^{NGA-RA}$ | 0.910, 0.925, 0.869, 0.918 | - | 0.698, 0.710, 0.687, 0.704 | - | 0.780, 0.760, 0.731, 0.780 |
|  | Proposed $Tf^{NGA-RA}$ | 0.900, 0.937, 0.875, 0.931 | - | **0.710, 0.720, 0.691, 0.749** | - | **0.783, 0.768, 0.739, 0.789** |

**Note:**
For a task $C_1$ *vs.* $C_2$, each entry denotes the accuracy, area under the curve, sensitivity, and specificity of the corresponding model respectively. The bold entries indicate the highest performance.

ParaHippocampal_L), (Amygdala_L, ParaHippocampal_R), (Amygdala_R, Hippocampus_L), (Putamen_L, Temporal_Inf_L), (Putamen_L, Amygdala_L), (Putamen_R, ParaHippocampal_L), (Putamen_L, ParaHippocampal_R), (Putamen_L, Hippocampus_L), (Putamen_L, Temporal_Inf_L), (ParaHippocampal_L, Hippocampus_L), (ParaHippocampal_L, Temporal_Mid_L), (ParaHippocampal_R, Amygdala_L), (ParaHippocampal_L, Temporal_Inf_L), (ParaHippocampal_R, Cuneus_L). These areas are also conformed by previous studies (*Vaithinathan, Parthiban & Initiative, 2019*; *Zhang et al., 2021*), which show strong connections to AD and MCI classes. Now, we compare the performances in differentiating various AD stages of our models with four other SOTA three-dimensional (3-D) deep learning models: 1. CNN3-D (*Dyrba et al., 2021*), 2. VGG3-D (*Korolev et al., 2017*), 3. ResNet3-D (*Korolev et al., 2017*).

Same training/test data is used as the inputs for the 3-D deep learning models (1–3) and the preprocessing pipeline is explained in "MRI Data and Pre-Processing" and are resized to $110 \times 110 \times 110$. Table 9 summarizes the comparison of performances of various three-dimensional (3-D) deep learning models with our method. The results furnished are the averaged accuracies of the models for $N = 25$ random runs with same train/test set. For our $Tf^{NGA-RA}$ model, only highest accuracies in various classifications are listed.

For AD *vs.* NC, all the models show an average of 94.5%. The proposed model ($Tf^{NGA-RA}$) performs better with an increase of 3–4% in MCI-based classifications than other models. Specifically, the $Tf^{Relu-RA}$ model comes close to the proposed $Tf^{NGA-RA}$ model. In the ADNI-1 training set, $Tf^{Relu-RA}$ performs better than $Tf^{NGA-RA}$ by a margin of 0.7–0.9% for AD *vs.* cMCI/MCInc. For all the other classifications, there is an increase in accuracy of 1–2% for $Tf^{NGA-RA}$ model than $Tf^{ReLU-RA}$ model.

The $Tf^{NGA-RA}$ performance was rigorously compared against four state-of-art models, especially CNN3-D, VGG3-D, ResNet-3D, and $Tf^{ReLU-RA}$, by paired t-Test based on their AUC values on different classification tasks. Figure 6 depicts the output

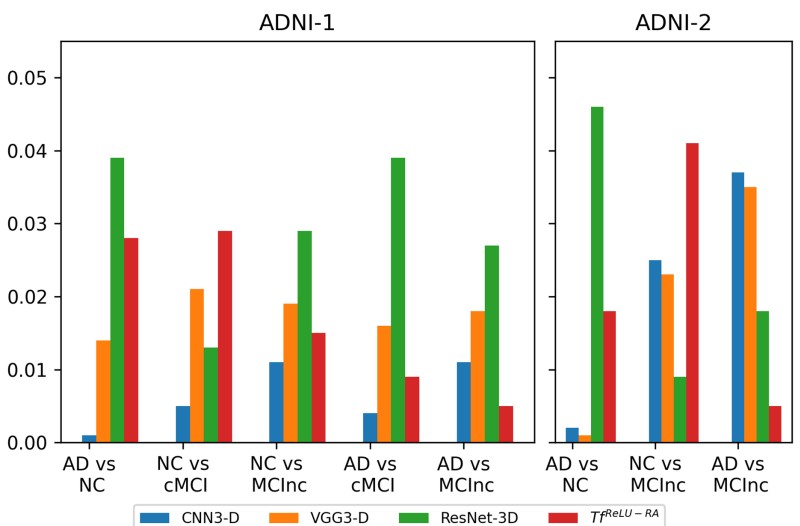

**Figure 6** The $Tf^{NGA-RA}$ performance is compared with CNN3-D, VGG3-D, ResNet-3D, and $Tf^{ReLU-RA}$, by paired $t$-Test with their AUC values on different classification tasks using the ADNI-1 & ADNI-2 datasets.                               

graphically. All results show that the difference in almost all tasks was statistically significant ($p$ ¡ 0.05).

For ADNI-1 training dataset, in the classification task of AD $vs$. NC, the $Tf^{NGA-RA}$ showed statistically significant improvements compared to all of the four models: CNN3-D, $p = 0.001$; VGG3-D, $p = 0.014$; ResNet-3D, $p = 0.039$; and TF, $p = 0.028$. This means that the proposed model could distinguish Alzheimer's from normal controls more effectively, a task extremely important for early diagnosis. For the NC $vs$. cMCI task, in which the focus was detection of normal controls $vs$. converters to mild cognitive impairment, the $Tf^{NGA-RA}$ again came up with statistically significant differences from CNN3-D ($p = 0.005$), VGG3-D ($p = 0.021$), ResNet3D ($p = 0.013$), and TF ($p = 0.029$), thus outperforming in the identification of this at-risk population. Trends from the NC $vs$. MCInc and AD $vs$. cMCI were similar with $Tf^{NGA-RA}$ significantly outperforming the older models in AUC performance, $p$-values ¡ 0.05 for all tests. Significant improvements are also reported for AD $vs$. MCInc with $p$-values: 0.011, 0.018, 0.027 and 0.005 against CNN3-D, VGG3-D, ResNet-3D and $Tf^{ReLU-RA}$, respectively.

In ADNI-2 training dataset, for the AD $vs$. NC classification task, it performed better than CNN3-D ($p = 0.002$), VGG3-D ($p = 0.001$), ResNet-3D ($p = 0.046$), and TF ($p = 0.018$). The results obtained show that the $Tf^{NGA-RA}$ is able to classify the AD and NC groups better and, hence, the model is quite helpful in early diagnoses where the most important thing in treatment planning is proper classification. This is complemented by NC $vs$. MCInc in which normal controls are compared with non-converters to mild cognitive impairment, where $Tf^{NGA-RA}$ also achieved statistically significant improvements. Better than CNN3-D at $p = 0.025$, VGG3-D at $p = 0.023$, ResNet-3D at $p = 0.009$, and TF at $p = 0.041$, it shows the $Tf^{NGA-RA}$ is more sensitive to the subtle

differences between the two conditions and necessary for tracking disease progression. The $Tf^{NGA-RA}$ outperforms the other models on the task AD $vs.$ MCInc, and the superiority is highly significant compared to CNN3-D by $p = 0.037$, VGG3-D by $p = 0.035$, ResNet-3D by $p = 0.018$, and TF by $p = 0.005$. In that respect, the model suggests an important ability to distinguish between AD and nonconverter MCI-a big challenge in the classification of early-stage disease.

Overall, the $Tf^{NGA-RA}$ consistently showed better performance in all the tasks mentioned above with statistically significant improvements in AUC, thereby supporting the idea that a clinical conclusion could be drawn accurately for the stage of Alzheimer's disease with the proposed model against traditional models.

To conclude, the features obtained from the proposed normalized group activations feature extraction networks give better results for AD-based classifications using MRI data. From the results, it is highlighted that the combination of activations can improve the performance of multi-stage AD classifications.

## LIMITATIONS AND FUTURE SCOPE OF WORK

This study on Alzheimer's disease classification using normalized group activations presents promising features, but a few limitations are worth mentioning. ADNI is a well-characterized dataset, but as a single cohort, it may not be fully representative of heterogeneous real-world clinical data. In our experiments, we are at risk of overfitting to the ADNI-specific population, and hence this model is not guaranteed to generalize well to other datasets or clinical settings. The next step is to validate the model using external data sets (*e.g.*, OASIS, AIBL or real clinical hospital data) to further examine its capabilities across populations and under clinical settings.

The existing model is mainly tailored towards to the structural and resting-state fMRI regimes. While in AD research, these imaging modalities are heavily used the model does not include other modalities *i.e.*, PET scans, CT scans or genomic data which have been shown to be useful. For further work, it would be crucial to combine PET with CT and other data such as EEG recordings or even genomic or proteomic data to get a better understanding of the model in terms of accuracy and robustness. This will also allow the model to find additional biomarkers linked with AD progression.

The model itself, especially by using transformer based architecture and group normalization activation functions, consumes heavy computational resources: high-performance GPUs (*e.g.*, RTX 3080) and large amount of memory. One direction for further work is to apply model compression techniques, knowledge distillation, quantization or pruning, on the models to make the model less complex and memory-efficient but keep their performance. A second strategy could involve deploying cloud-based solutions which would perform computationally demanding tasks on cloud servers thereby allowing model predictions to be accessed by clinicians who may not have the extensive hardware resources needed.

Even though there have been advances in model interpretability, "black box" perception of deep learning models is common specially in healthcare. Clinical professionals need more insight into how these decisions are made, especially in the diagnosis of such serious

diseases as AD. In the future, linking to explainable A.I. (XAI) methods may be interesting such as LIME or SHAP, in which we can decompose the decision-making path of our model, and thus make it clearer for clinicians to comprehend why certain predictions were made by the model. Moreover, to reduce the performance-trust gap of the model, we can work with healthcare professionals directly to co-design interpretability so the model is more reliable for healthcare managers.

Enriched feature extraction, such as scattering transforms, wavelet features, and GLCM textures are applied more frequently which may prone to overfitting especially when dealing with a small sample size for deep learning models. Future directions might concern models or add-ons tailored to early pre-clinical diagnosis including longitudinal information from biosignatures, genetics, and cognitive tests. By developing models that capture progression from healthy to MCI to AD over time, we believe we can build a more holistic view of early intervention. In the future work, the regularization methods of the model such as dropout, data augmentation and cross-validation strategies should be improved to reduce the risk of overfitting. One alternative measure to minimize this problem could be a boost in data using data augmentation, or integrating external databases.

## CONCLUSION

In this study, normalized group activations based feature extraction technique using both structural MRI and resting-State-fMRI is developed and evaluated on heterogeneous data. For this purpose, an automated AD diagnosis system is designed and implemented on multi-cohort ADNI data to predict multi-stages of AD. To illustrate the good visual representation of the proposed features, connectograms of rs-fMRI functional connectivities of top discriminatory regional pairs for AD $vs.$ NC, $e_{MCI}$ $vs.$ $l_{MCI}$, NC $vs.$ $e_{MCI}$ and AD/MCI-specific rollout attention explainability with transformers are also graphically shown. In all classification tasks, the non-transformed features, wavelets, and curvelets perform better than other features in both modalities. Results indicate that the combined activational features of linear rectifier, exponential linear, and exponential linear sigmoid squashing functions yield good results for MCI classifications. This proves that our novel group activation features are suitable for baseline rs-fMRI time-series and MRI data. Among the transformed features, wavelets show the potential biomarker characteristic with good AD/MCI predictions. The high memory/time complexities and a way to find an optimized trade-off between the performance/interpretability with number of activations is a major challenge. One solution is to reduce the processed regional pairs using genetic, evolutionary, and other bio-inspired algorithms. Another way is to improve our model to adapt in the compositional learning environment, $i.e.$, the features of the singular disease can be trained in a distributed way which provides faster runtime and enhanced performance. Also, spatial and graph representation learning can be applied onto this context to obtain in-depth understanding about the feature interpretability. It is noted that our technique have broader scope and will be usefull to the researchers in this field. Furthermore, to the best of our knowledge, it is highlighted that this is the first study

to investigate group activations on multiple transformed features for both binary and ternary AD classifications.

## ACKNOWLEDGEMENTS

This project was made possible through publicly available data provided by the Alzheimer's Disease Neuroimaging Initiative (ADNI) (National Institutes of Health Grant U01 AG024904) and DOD ADNI (Department of Defense award number W81XWH-12-2-0012).

### Funding

This research was funded by Taif University, Saudi Arabia, Project No. (TU-DSPP-2024-52). The funders had no role in study design, data collection and analysis, decision to publish, or preparation of the manuscript.

### Grant Disclosures

The following grant information was disclosed by the authors:
Taif University, Saudi Arabia: TU-DSPP-2024-52.

### Competing Interests

The authors declare that they have no competing interests.

### Author Contributions

- Krishnakumar Vaithianathan conceived and designed the experiments, analyzed the data, performed the computation work, prepared figures and/or tables, and approved the final draft.
- Julian Benadit Pernabas conceived and designed the experiments, authored or reviewed drafts of the article, and approved the final draft.
- Latha Parthiban performed the experiments, authored or reviewed drafts of the article, and approved the final draft.
- Mamoon Rashid performed the computation work, authored or reviewed drafts of the article, and approved the final draft.
- Sultan S. Alshamrani analyzed the data, prepared figures and/or tables, and approved the final draft.

### Data Availability

The code is available at GitHub and Zenodo:

- https://github.com/vkichu77/ADClassificationWithHetergenousData

- Krishnakumar Vaithianathan. (2024). vkichu77/ADClassificationWithHetergenousData: Code Submission 1 (v1.0.0). Zenodo. https://doi.org/10.5281/zenodo.13906540

The ADNI Data is available at: https://adni.loni.usc.edu/data-samples/adni-data/#AccessData.

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
