# Peer review of "Normalized group activations based feature extraction technique using heterogeneous data for Alzheimer’s disease classification"

_PeerJ Computer Science, doi:10.7717/peerj-cs.2502_

## Round 0.1 · original submission · Major Revisions

The authors shall address the reviewers' queries to satisfaction. In addition, the authors shall elaborate on the novelty of the proposal, the computational requirements and comparison to existing literature, and add statistical significance analysis to demonstrate that the improvements in performance are not by chance. The questions on the selection of feature extraction methods and methods of model optimization need to be properly addressed. The limitations of the proposal and scope for future work shall be discussed in detail.

·

Basic reporting

No comment

Experimental design

No comment

Validity of the findings

1) Give a more detailed description for images in the dataset
2) Define each term in all equations
3) Add a detailed comparison (in tabulated form) with previous work in the field to show the significance of your results.

Additional comments

No additional comments

Reviewer 2 ·

Basic reporting

Paper is about feature extraction technique based on normalized group activations that can be applied to both structural MRI . An automated diagnosis system equipped with
the proposed feature extraction is designed and analyzed on multi-cohort Alzheimer's
Disease Neuroimaging Initiative(ADNI) data to predict multi-stages of AD. author should give more justification on the proposed feature extraction technique's. Till date many extraction techniques are available then how your is novel than the existing.

Experimental design

Include the column names in Table 5.
In Table 9: Comparison of recent methods, authors inserted results in bold text. Author should elaborate the importance for the same. some existing method (ResNet3-D) results are better than proposed method, if it is correct then justify proposed method novelty.

Validity of the findings

No comments

Additional comments

Paper sir well written. Author should give more focus on following points
1. Author should give more on novelty of the proposed features extraction model how it is better than FCNet(Riaz et al., 2020)
2. Alexnet and ResNet are normally extract features automatically then what is use of pass extracted features through this models. Give the proper justification.

·

Basic reporting

The authors done a good job.
English is good
Given all the relevant information.
Abstract is good and it matches the conclusion.

Experimental design

Good
All the relevant information are give for this work

Validity of the findings

Given correct explanation for everything.

Additional comments

The way of writing is good.
Done enough survey and given in literature.
Good job.

Reviewer 4 ·

Basic reporting

The study introduces a unique feature extraction technique based on normalized group activations, which is applicable to both structural MRI and resting-state fMRI data.

Experimental design

The results provided should be substantiated with publicly available code. Provide github links for the code.

Validity of the findings

The proposed technique is applied to both rs-fMRI and MRI data; however, it briefly mentions the use of non-transformed features, curvelets, wavelets, and other techniques. It would be valuable to include a detailed comparative analysis of the proposed method against these established feature extraction methods to better quantify its advantages and limitations, and to provide insights into its relative strengths in different neuroimaging modalities. It would be better to conduct ablation study to understand fundamental benefits/limitations of these features.

Additional comments

The paper is well written however if it includes the following, it would further enhance the quality:

1) Providing publicly available code with appropriate comments for ease of checking by the research community.

2) Conducting ablation study.

---

## Round 0.2 · accepted · Accept

The authors have addressed the reviewers' comments to satisfaction.